



# Response of water balance components to climate change in permanent grassland soil ecosystems

Veronika Forstner[1*], Jannis Groh[2,3*], Matevz Vremec[1], Markus Herndl[4], Harry Vereecken[3], Horst H. Gerke[2], Steffen Birk[1], Thomas Pütz[3]

[1]Institute of Earth Sciences, NAWI Graz Geocenter, University of Graz, Graz, 8010, Austria
[2]Working Group "Hydropedology", Research Area 1 "Landscape Functioning, Leibniz Centre for Agricultural Landscape Research (ZALF), Müncheberg, 15374, Germany
[3]Institute of Bio- and Geoscience IBG-3: Agrosphere, Forschungszentrum Jülich GmbH, Jülich, 52425, Germany
[4]Institute of Plant Production and Cultural Landscape, Agricultural Research and Education Centre, Irdning-Donnersbachtal, 8952, Austria

* These authors contributed equally to this work.
*Correspondence to*: Veronika Forstner, Jannis Groh (veronika.forstner@uni-graz.at, j.groh@fz-juelich.de or groh@zalf.de)

## Abstract

Hydrological processes are affected by changing climatic conditions. In grassland areas, changes in the ecosystem water balance components will alter aboveground biomass production (AGB), which in turn is of great importance for ecological and economic benefits of grassland. However, the effects of climate change on the ecosystem productivity and water fluxes are often derived from climate change experiments. It is still largely unknown whether and how the experimental approach itself affects the results of such studies. The aim of this investigation was to identify the effects of climate change on the water balance and the productivity of grassland ecosystems by comparing results of two contrasting approaches of climate change experiments. The first (manipulative) climate change approach uses increased atmospheric $CO_2$ concentrations and surface temperatures. The second (observational) approach uses data from a space-for-time substitution approach along a gradient in climatic conditions. The climate change effects on the ecosystem's water balance was determined by using high-precision weighable monolithically lysimeters at each site over a period of four years, including the exceptionally dry year 2018. The aridity index, defined as the grass-reference evapotranspiration ($ET_0$) to precipitation (P), was used to characterize the hydrological status of the regime (i.e. energy- or water limited system).

The observational approach (grassland ecosystem moved to a drier and warmer site), resulted in a large decrease of precipitation (P) and non-rainfall water (NRW), an increase in actual evapotranspiration ($ET_a$) and upward directed water fluxes from deeper soil and hence a decline of seepage water as well a decrease in AGB and water use efficiency (WUE). The manipulative approach (grassland ecosystem treated in place) resulted in decreasing P and NRW under conditions of elevated temperature but responded with increasing NRW for elevated $CO_2$ as compared to the reference. Similarly, an elevated $CO_2$



and heating increased the ecosystem's water loss by $ET_a$. However, the effect of increasing $CO_2$ on $ET_a$ was largely compensated by the opposite effect of an elevated temperature in the combined treatment. The seepage water rate also increased

with elevated $CO_2$, whereas it clearly decreased for the heating treatment as compared to the reference. All treatments led to a reduction of the grassland productivity in terms of the AGB and to reduced WUE as compared to the grassland ecosystem under reference conditions.

The consideration of changes in NRW and P by the treatments needs to be considered in climate change experiments to avoid an over- (elevated temperature) or underestimation (elevated $CO_2$) of the effects of climate change on ecosystems response,

especially for sites where water limitation plays a role. The impact of drought periods on seepage rates (potentially leading to groundwater recharge) was more pronounced for the relatively humid site with a longer $ET_a$ period without water stress than for a relatively dry site. The hydroclimatological and ecohydrological indicators were similarly affected by changes in temperature, atmospheric $CO_2$ concentrations, and precipitation in both manipulative and observational climate change experiments except for the responses of $ET_a$ and AGB in the dry and warm year 2018. The resulting response differences

between the two climate change approaches were explained by the actual soil moisture status. The results suggest that energy limited ecosystems tend to increase their $ET_a$ and AGB production (excluding effects from elevated $CO_2$ and temperature), but water limited ecosystems respond with a decrease in $ET_a$ as a result of water stress, which leads to a clear decline of AGB.

The results also suggest that climate change experiments should account for the possible change of the hydrological status of the ecosystem and impose sufficiently extreme levels of climatic conditions within their set-up to allow such changes to occur

for capturing the full response of the ecosystem. The results may help to better understanding the impact of climate change on future ecosystem functioning.

## 1 Introduction

Current and future climate change is expected to alter air temperature, $CO_2$ concentration in the atmosphere as well precipitation (Abbott et al., 2019; IPCC, 2018). Changes in these conditions will alter hydrological processes and affect the

soil water availability, which is of critical importance for the agricultural sector in terms of plant development and food production (Thornton et al., 2014). Grassland is one of the Earth's most important biomes (Blair et al., 2014). Managed grassland areas are important for carrying capacity of livestock as well herbage and hay production for forage. Moreover, these areas are also important for several other ecosystem services beside food production like water supply and flow regulation, erosion control, climate mitigation, pollination, carbon storage as well as cultural services (Bengtsson et al., 2019). All these

services are strongly dependent on weather and climate conditions and thus potentially highly vulnerable to climate change (Gobiet et al., 2014). Since the late 19[th] century, air temperatures in the Alpine region have risen about twice as much as the global or Northern Hemispheric average (Auer et al., 2007). These strongly changing climatic conditions, in particular the expected increase in frequency and magnitude of extreme events such as droughts and heavy rainfalls, are likely have adverse effects on the soil water balance of grassland ecosystem and/or adverse or possibly beneficial effects on grassland biomass





production, especially in mountainous regions. Depending on the regional climate change and its local impacts, altered precipitation regimes and higher temperatures potentially will increase evapotranspiration and thus negatively affect the local ecosystem services related to water (Schirpke et al., 2017; Rahmati et al., 2020). The influence of droughts and higher temperature on grassland biomass production is less clear but a shift in the temperature regime will prolong the growing season, which might change the vegetation composition, water use efficiency (WUE) of grasslands, enable a more intensified use of

grassland sites (more frequent mowing) and increase the biomass production (Eitzinger et al., 2009; Tello-García et al., 2020). For regions that are currently disadvantaged because of the climatic and topographic conditions, it may be expected that in the future biomass production will increase due to higher temperatures (Eitzinger et al., 2009). However, the combined effect of higher temperatures, causing increased evapotranspiration and the expected decrease in summer precipitation also suggests more frequent and more severe occurrences of droughts, which may lead to plant water stress and thus adversely affect the

aboveground biomass (AGB) production of grassland as well as the quantity and quality of the drainage water (Herndl et al., 2019).

To contribute to the assessment of climate change impacts on mountain grassland, there is the general need to understand the individual and combined effects of changes in temperature and precipitation with and without elevated atmospheric $CO_2$ concentrations on the water balance components and the connected biomass production of mountain grasslands. This implies

precipitation (P), the formation of non-rainfall water (NRW; i.e. dew and fog), evapotranspiration ($ET_a$), seepage water as well the ecosystem productivity (i.e. AGB), in particular during droughts. The use of high precision weighable lysimeter allows quantifying the water balance components of ecosystems and determining their productivity, e.g. shown in Groh et al. (2020), but requires a thoroughly processing of lysimeter data by applying quality checks and data filter (Peters et al., 2017; Groh et al., 2019). It may be expected that even for northern humid ecosystems NRW formation temporarily gains importance in the

water budget during droughts (Groh et al., 2018) and that under such extreme conditions, heat and/or water stress can cause a decrease in the AGB production of grasslands (Fu et al., 2006). To explore how ecosystems will respond to changes in environmental conditions key hydroclimatical and ecohydrological indicators such as the precipitation use efficiency (PUE; Wang et al., 2019), WUE (Hatfield and Dold, 2019; Groh et al., 2020), and runoff coefficient (Chen et al., 2007) can be used to assess the impact of changing climatic conditions on the ecosystems.

To address these research questions, data from either manipulative experiments (using multifactorial drivers) or observational experiments on environmental gradients are one major tool to explore the relationship between changing climate factors and ecosystem responses (Hanson and Walker, 2020; Song et al., 2019; Kreyling and Beier, 2013; Knapp et al., 2018). According to Yuan et al. (2017), the main difference between these approaches is the issue of association versus causality, as observational approaches can identify a relationship between a set of randomly selected variables of a system under natural conditions, while

manipulative approaches are more likely to identify and confirm the underlying mechanism based on measuring responses of the system by controlling certain variables. It is often assumed that manipulative or observational climate change experiments lead to similar results in order to assess the effects of climate change on e.g. the components of the water balance. However, a





recent meta-analysis on climate change experiments showed that the impacts of climate change on the nutrient cycle differed among the tested approach (Yuan et al., 2017). Knapp et al. (2018) compared the response of different grassland ecosystem

on climate change from manipulative and observational climate change experiments and found that both approaches achieved similar functional relationship between growing season P and NRW (GSP) and AGB, when the growing season precipitation was within the range of historical observations. The predictions outside the range of historic events lead to non-linear relations in the AGB response versus the change in growing season precipitation ( Knapp et al., 2018). This clearly demonstrates the need to impose relatively extreme changes in the boundary conditions (e.g. precipitation reduction) in climate change

experiments in order to observe the non-linear response of the soil-ecosystem (i.e. AGB) to changes in the climate regime (Knapp et al., 2018). We therefore hypothesize that the response of the grassland ecosystems on changing climatic conditions will differ among the two approaches. The main objective of the present study was to test this hypotheses by comparing the impact of climate change on (i) the water balance and (ii) the AGB production within and between the manipulative and observational approach, and (iii) to identify the impact of altered climate conditions on the functional relationships between

water balance components, AGB, and hydroclimatical and ecohydrological indicators. Data are provided from two distinct climate change experiment, namely "Lysi-T-FACE" (Climate Impact Research on Grassland; Herndl et al., 2010; Herndl et al., 2011) and TERENO-SOILCan (TERrestrial ENvironmental Observatories; Pütz et al., 2016). The Lysi-T-FACE experiment was defined here as manipulative climate change approach and the TERENO-SOILCan as an observational. In both experiments, high precision weighing lysimeters were used to quantify the water balance components and the AGB of

grassland ecosystem in mountainous regions.

## 2 Materials and Methods

### 2.1 Manipulative and observational approaches

The climate change experiment Lysi-T-FACE is an experimental concept that has been designed to enable warming grassland plots using an infrared heating system (Kimball et al., 2008), and enriching the $CO_2$ content of the air using a Mini-FACE

system (T-FACE; Miglietta et al., 2001). This experimental set-up was implemented at an alpine grassland site in 2010 and 2011 (Herndl, 2011). The overall experimental design at this site is based on a surface response approach (Piepho et al., 2017) and includes factor combinations of two elevated temperatures and two elevated $CO_2$ concentrations at 24 grassland plots under open-field conditions. The Lysi-T-FACE approach at weighable lysimeters (Fank and Unold, 2007) is implemented at six of these plots. The proper functioning of the T-FACE performance has been repeatedly tested and is fully operated since

May 2014 at the experimental site.

The observational approach for quantifying climate change impacts in the soil-plant system is implemented in the TERENO-SOILCan lysimeter network since 2010 / 2011 (Pütz et al., 2016; Pütz et al., 2018). Intact soil monoliths were transferred within and between TERENO observatories to expose them, apart from the observations at their original site, to other climatic





conditions (space-for-time substitution; see details in Pütz et al., 2016; Groh et al., 2020). The concept of the space-for-time
substitution means that soils were translocated in space instead of waiting at the same location for changes in climatic
conditions in time. The change of the climate regime by the transfer of the lysimeters were abrupt, which implies that we are
not able to detect gradual changes of the grassland ecosystem over time as suggested in standard space-for-time approach, but
can account for unsuspected effects from the past (Groh et al., 2020).

## 2.2 Lysimeter set up

The study was conducted at three test sites (Table 1). The Lysi-T-FACE test site is located at the Agricultural Research Centre
Raumberg-Gumpenstein (GS) in Austria. The experimental site is located at an altitude of 707 m a.s.l. within the Enns valley
of the Austrian Alps. The mean air temperature at the site is 7.2 °C, mean annual precipitation is 1000 mm; and the soil is a
Cambisol. Thus, the site may be considered as representative of permanent grassland in the alps (Schaumberger, 2011). The
grasses *Arrenatherum elatius* and *Festuca pratensis* and the leguminous species *Lotus corniculatus* and *Trifolium pratense*
dominate the grassland established at the Lysi-T-FACE site. The grassland was mowed three times per year (see supplement
Table S 1), each followed by mineral fertilization (Herndl et al., 2010).

The two other test sites are part of the TERENO-SOILCan lysimeter network, are located in the Northwest of Germany, in
Rollesbroich (RO) and Selhausen (SE). Both sites have a humid temperate climate with an average annual air temperature of
8 °C and 10 °C and an average annual precipitation of 1150 mm and 720 mm for Rollesbroich and Selhausen, respectively.
145   The plant community consists mainly of *Lolium perenne* and *Trifolium repens*. The grassland lysimeters at both sites were
subject to management (cutting and fertilizer) according to the local agricultural management of the surrounding field at
Rollesbroich. This includes three to four cuts per growing season (supplement Table S1) and three to four times application of
liquid manure or mineral fertilizer per year (Pütz et al., 2016).

The three test fields are equipped with high precision weighable lysimeters with the similar design (METER Group). At GS
150   six lysimeters are installed; one lysimeter is operated under ambient conditions (C0T0), two under a constant warming of 3 °C
of grassland (C0T2) relative to the ambient surface temperature, two under a constant elevated $CO_2$ concentration (300 ppm)
relative to the ambient atmospheric $CO_2$ concentration (C2T0), and one with a combination of elevated temperature and
elevated $CO_2$ (C2T2). The used abbreviations C and T within the treatments stand for $CO_2$ and temperature and the numbers
0 for ambient and 2 for elevated conditions. A mechanical snow cover separation system (METER Group) is used to maintain
a correct water balance in winter months. Manual snowpack separation is only required in periods with a very high snowpack.
A nearby weather station was used to obtain meteorological observations of reference precipitation (tipping-bucket method;
Young GWU), air temperature, air humidity, solar radiation, net radiation and wind speed at a height of 2 m above ground.
Data from these stations was used to calculate grass-reference ET according to Penman-Monteith (Groh et al., 2019). At RO
(C0CL0, CL stands for climate) six lysimeters and at SE (C0CL2) three lysimeters from RO were installed to quantify soil



water budget under ambient (RO: C0CL0) and altered (SE: C0CL2, less precipitation, higher potential ET) climatic conditions. A weather station (WXT510, Vaisala Oyj) was installed at both sites logging the same meteorological parameters as at GS. The reference precipitation was measured with a weighing rain gauge (OTT Pluvio², OTT HydroMet GmbH) and a net radiation sensor (LP Net07, Delta OHM S.rL.) was installed above one lysimeter at each site. In addition, vegetation height observations were obtained for calculating the grass-reference ET with Penman-Monteith model (Allen et al., 2006).

**Table 1:** Overview of the three test sites with their different approaches at the location Gumpenstein (GS), Rollesbroich (RO), and Selhausen (SE). The experiments at GS comprise treatments that have ambient (C0T0), elevated air temperature (T) (C0T2), elevated concentration of $CO_2$ (C2T0), and combined elevated concentration of $CO_2$ and T (C2T2) conditions. In the observational space-for-time substitution approach, site RO represents ambient atmospheric demand for evapotranspiration and ambient precipitation (C0CL0), while the site SE represents elevated atmospheric demand for evapotranspiration and reduce amount of precipitation (C0CL2).

| Site | Project | Approach | Treatment | Symbol |
|---|---|---|---|---|
| Gumpenstein (GS): 707 m.a.s.l | Lysi-T-FACE | Manipulative: Lysi-T-FACE | ambient<br>+3.0 °C<br>+300 ppm $CO_2$<br>+300 ppm $CO_2$ ; +3.0°C | C0T0<br>C0T2<br>C2T0<br>C2T2 |
| Rollesbroich (RO): 511 m.a.s.l | TERENO-SOILCan | Observational: Space-for-time | Original | C0CL0 |
| Selhausen (SE): 104 m.a.s.l | TERENO-SOILCan | Observational: Space-for-time | Translocated | C0CL2 |


## 2.3 Quantifying water balance components

Weighable high precision lysimeter systems provide measurements on the components of the soil water balance equation, P, NRW, $ET_a$, and the vertical net flux above the lysimeter bottom (NetQ). The NetQ component comprises water flow out (Q) and into (CR) the lysimeter at the bottom; note that CR mimics upward flow by the capillary rise. The change in soil waters

storage (ΔS) was calculated as:

$$\Delta S = P + NRW - ET_a - Q + CR \tag{1}$$

The water balance components P and $ET_a$ were obtained at each site from the highly resolved (1-minute) and precise (resolution: 0.01 mm) lysimeter observations over a period of four consecutive years (2015-01-01 until 2018-12-31). The water flux across the bottom boundary of the lysimeter (NetQ) was obtained at the same time interval from a weighable water tank

(resolution: 0.001 mm). Lysimeter mass changes are prone to external disturbances like management operations or wind. Thus, lysimeter data of mass changes have been processed by pre- and post-processing routine to avoid that external errors and noise affect the determination and separation of water fluxes across the land surface (P, NRW and $ET_a$). The procedure included a visual data quality check of the one-minute lysimeter data as well as an application of the Adaptive Window Adaptive



Threshold filter (Peters et al., 2017), which has been shown to allow the quantification of small water fluxes like dew, water
vapour adsorption or nighttime ET (Groh et al., 2019; Kohfahl et al., 2019). Water fluxes across the land surface and bottom
boundary were aggregated to 10-min time intervals for detailed analysis of precipitation, especially the identification of NRW
(Groh et al., 2018), and $ET_a$. Gaps in the time series of measured precipitation and evapotranspiration were filled with
observations from either from parallel lysimeter observations or from the external rain gauge or in terms of calculated grass-
reference ET ($ET_0$) with the Penman-Monteith model (Allen et al., 2006) using the meteorological data for hourly time steps.
Finally, all water balance components were aggregated on a daily time scale and were averaged over the number of available
repetitions for each treatment. At GS the water balance components for the treatment C2T0 in 2017 and 2018 were taken only
from one lysimeter due to technical problems.

The component NRW, which includes fog, dew and hoar frost formation, was determined by lysimeter mass increases between
sunset and sunrise, when the corresponding rain gauge from the meteorological station did not detect precipitation during the
corresponding 10 minute time step (Groh et al., 2018). The treatment of the Lysi-T-FACE plots at GS, i.e. free air carbon
enrichment and infrared heating together, was active only during the growing season when active plant growth occurs. In
winter periods, $CO_2$ enrichment is totally out of operation. It should be noted that the $CO_2$ enrichment in the manipulative
approaches (C2T0 and C2T2) is deactivated when soil temperature in 10 cm is below 3 °C; a condition as used in the set-up
at GS to define the non-growing season. However, in this work, the non-growing season is defined in a different way as
described next for comparing the manipulative approach with the observational approach. In contrast to the $CO_2$ enrichment,
the heating is not totally out of operation in winter periods, heating is only turned off if the snow cover is higher than 10 cm.
Thus, comparison of the two treatments, free air carbon enrichment and infrared heating, is possible only in the growing season.
Hence the water balance components of the different soils at the specific sites under the original and climate change conditions
were compared separately for the growing season and non-growing season. As a quantitative measure of the effect of changes
in the climate variables, the mean error representing the average deviation between the daily values obtained under changed
conditions and those of the ambient reference is calculated. The R software (R-Core-Team, 2016) and the function *me* of the
package hydroGOF (Zambrano-Bigiarini, 2017) were used to calculate the mean error as:

$$Mean\ error = \frac{1}{N}\sum_{i=1}^{N}\left(Obs_{Ti} - Obs_{refi}\right) \hspace{4cm} (2)$$

where n is the number of samples and $Obs_{Ti}$ and $Obs_{refi}$ are the daily value on the corresponding water balance term from the
treatment (subscript Ti) and reference (subscript refi). The AGB production of the grassland ecosystem were analysed solely
for the growing season. A thermal based definition of the growing season and non-growing season proposed by Ernst and
Loeper (1976) was employed in this study to identify the beginning and ending of the growing season /non-growing season at
the corresponding site. In this approach the beginning of the growing season can be obtained by adding all positive average
daily air temperatures from January 1st up and considering specific weight factors for each month. The daily means of air
temperatures are multiplied by weighting factors of 0.5 (January), 0.75 (February), and 1 (March) and added. The beginning



of the growing season in spring was defined as the day when the cumulative temperature sum exceeded a threshold of 200° C. The same approach is used to obtain the end of the growing season, yet starting the temperature sums backward from December 31, with weighting factors of 0.5 and 0.75 for December and November, respectively.

**2.4 Hydroclimatological and ecohydrological indicators**

The dry matter AGB was gravimetrically determined with a precision balance (Gumpenstein: EA 6DCE-I, Sartorius; Selhausen and Rollesbroich: EMS 6K0.1, KERN) after drying at 55°C for 48 hours for dry AGB at GS and 60°C for 24 hours at RO and SE. The AGB from the different cuts were summed over the growing season to annual values for each test site and averaged over the replications were calculated to compare the productive between the different treatments and climate change approaches (see Table 1).

The crop WUE, defined here as the amount of dry AGB produced per unit of water used by a plant (Tello-García et al., 2020), was estimated as:

$$WUE = \frac{AGB}{ET_a} \tag{3}$$

where AGB represent the dry matter biomass production (g m$^{-2}$) and $ET_a$ the actual evapotranspiration (mm) during the growing season of the corresponding year, both of which were obtained by averaging over the lysimeters with the same
treatments. The annual PUE (g m$^{-2}$ mm$^{-1}$), which are defined as the ratio of AGB and mean annual P (Zhou et al., 2020) and NRW was calculated as:

$$PUE = \frac{AGB}{P+NRW} \tag{4}$$

The PUE is a key indicator that explains the response of the ecosystem productivity to P (Wang et al., 2019) and NRW. Hence, PUE is used here to explore how the relationship between water balance and the crop components in the carbon cycle reacts
to changes in the environmental conditions (Zhou et al., 2020). The aridity index (mm a$^{-1}$) and the ratio of seepage (Q) to P and NRW (-) were determined by equations (5) and (6):

$$AI = \frac{ET_0}{P+NRW} \tag{5}$$

$$QP = \frac{Q}{P+NRW} \tag{6}$$

The QP ratio is a dimensionless indicator that describes the portion of P and NRW that becomes seepage that eventually will
contribute to groundwater recharge. Here, we used QP ratio as indicator to assess how changing climate conditions affect the hydrological functioning of the similar soil-ecosystem. It should be noted that for QP, AI, and PUE normally only P is used, as quantitative information on NRW are often not available. Linear correlations between AGB, $ET_a$, $ET_0$, WUE, GSP, PUE, and AI were generated by the use of the package *lm* (R-Core-Team, 2016) to determine the responses of the corresponding





variable/indicator under changing climate and the reference conditions for manipulative and observational approach. The

significance level (p < 0.01) was used to indicate if the relationship between the variables are statically significant.

## 3 Results and discussion

### 3.1 Impact on water balance components

#### 3.1.1 Precipitation and non-rainfall water

The average annual P for the manipulative climate change approach at GS ranged from 1088 to 1131 mm $a^{-1}$ across the

observation period (2015 – 2018; Table 2; more details in Table S 3) and for the treatments. The P values were generally larger
during growing (average range: 691 to 724 mm) than non-growing seasons (average range: 396 to 408 mm; Table 2). The
average annual NRW ranged between 52 and 76 mm (Table 2; more details in Table S 4), which corresponds to 4.8 to 6.7 %
of the total annual P (Table S 5). In contrast to P, NRW was on average larger during the non-growing season (34 to 48 mm)
than during the growing seasons (20 to 28 mm), which agrees well with previous findings on the seasonality of dew formation

across different climate zones (Zhang et al., 2019; Groh et al., 2019; Atashi et al., 2019). The treatments at GS showed
differences in P and NRW, despite no active direct control on these variables. The lysimeter solely enriched with $CO_2$ (C2T0:
1131 mm) achieved on average similar annual P amounts over the observation period than the reference lysimeter (C0T0:
1125 mm). In contrast, the annual P amounts of the temperature-increased treatment (C0T2: 1096 mm) and the combined $CO_2$-
enriched and temperature-increased treatment (C2T2: 1088 mm) were on average smaller than the reference observation

260     (C0T0). A similar tendency on daily P between the treatments and the ambient measurement was observed for the non-growing
(Fig. 1 A) and growing season (Fig. 1 B), with negative mean error values for treatments C0T2 and C2T2 and positive values
for C2T0. Most pronounced differences between treatments with an elevated temperature and reference were detected during
the growing season and the mean error of daily P ranged between -0.12 to -0.14 mm $d^{-1}$ (Fig. 1 B). This demonstrates clearly
a warming-induced decrease in GSP (by up to 4 %) and NRW amounts (up to 24 %), which needs to be considered in global

265     change experiments to avoid overestimating the effects of global warming. This agrees well with Feng et al. (2021), who
showed that heating with an infrared heater system reduced dew formation for an alpine grassland ecosystem of the Tibetan
Plateau.

**Table 2:** Average values of the soil water balance components: precipitation (P), non-rainfall water (NRW), actual evapotranspiration ($ET_a$),

270     net water flux across the lysimeter bottom (NetQ), and change of the soil water storage (ΔS). The values were averaged across the non-
growing seasons, growing seasons and annual values for the period 2015 - 2018, from replicated lysimeter measurements at each test sites
Gumpenstein (GS) with C2T0 ($CO_2$: +300 ppm; two lysimeter), C0T2 (temperature: +3°C, two lysimeter), C0T0 (ambient, one lysimeter),



C2T2 (CO$_2$: +300ppm, and temperature: +3°C, one lysimeter), Rollesbroich (RO) with C0CL0 (original, six lysimeter) and Selhausen (SE) with C0CL2 (translocated, three lysimeter).

| Component | Season | Gumpenstein | | | | Rollesbroich | Selhausen |
|---|---|---|---|---|---|---|---|
| | | C2T0 | C0T2 | C0T0 | C2T2 | C0CL0 | C0CL2 |
| | | mm | | | | | |
| P | non-growing | 407.6 | 400.8 | 405.8 | 396.2 | 516.0 | 212.6 |
| | growing | 723.7 | 695.3 | 718.9 | 691.4 | 503.8 | 401.7 |
| | annual | 1131.3 | 1096.1 | 1124.7 | 1087.6 | 1019.8 | 614.3 |
| NRW | non-growing | 48.1 | 36.7 | 39.9 | 33.5 | 30.9 | 21.6 |
| | growing | 28.2 | 19.5 | 24.7 | 18.8 | 24.8 | 22.8 |
| | annual | 76.3 | 56.1 | 64.6 | 52.3 | 55.7 | 44.4 |
| ET$_a$ | non-growing | 114.9 | 133.7 | 109.6 | 125.0 | 87.3 | 88.7 |
| | growing | 541.6 | 696.8 | 620.2 | 645.0 | 552.0 | 596.1 |
| | annual | 656.5 | 830.5 | 729.8 | 770.0 | 639.3 | 684.7 |
| NetQ | non-growing | 351.5 | 272.3 | 326.0 | 285.3 | 429.4 | 30.0 |
| | growing | 201.4 | 54.5 | 122.1 | 61.9 | 28.3 | -58.2 |
| | annual | 552.9 | 326.8 | 448.0 | 347.2 | 457.7 | -28.2 |
| ΔS | non-growing | -10.7 | 31.5 | 10.1 | 19.4 | 30.2 | 115.5 |
| | growing | 8.9 | -36.5 | 1.3 | 3.3 | -51.7 | -113.4 |
| | annual | -1.8 | -5.1 | 11.5 | 22.7 | -21.5 | 2.2 |

All treatments had a visible impact on the formation of dew in comparison to the ambient conditions (C0T0; Table 2). The tendency of the different treatments on NRW was similar to P, with C2T0 values generally larger and C0T2 and C2T2 smaller compared to the reference. Observations reveal that an increase in canopy surface temperature (C0T2 or C2T2) goes along with a decreasing formation of NRW with negative mean error values during both periods (Fig. 1 E and F). The change in temperature and CO$_2$ concentrations reduced the annual NRW on average for C0T2 by 13 % and for C2T2 by 19 % in comparison to the ambient variant C0T0. Results for C2T2 indicates that the elevated surface temperature is the dominating factor in dew formation, as this treatment generally achieved the lowest relative contribution of dew formation on the total incoming water at the annual scale and the largest negative mean error values. Our results reveal that an elevated CO$_2$ concentration also seems to influence the formation of NRW, as the NRW increase by 18 % for treatment C2T0 compared to the reference C0T0. One reason might be that an increasing airflow above the canopy during the CO$_2$ application reduces the canopy temperature, increases the temperature gradient between canopy and air, thus enhancing dew formation. This agrees well with Feng et al. (2020), who reports that artificial warming of surfaces can reduce the amount of dew by up to 91 % in arid to semi-arid grassland ecosystems, which demonstrates that the effect of infrared heater warming system are an often overlooked factor in warming experiments. Not considering these effects of elevated warming and elevated CO$_2$ on NRW and P in manipulative climate change experiments may lead to an under- or overestimation of ecosystem responses to changes in P and NRW in climate change simulation studies.





Seasonal and annual P values and NRW amounts were generally significantly lower for 2018 compared to the other three years (Table S 3 and S 4). However, the percentage of NRW achieved on average with 5.7 % of the annual P a similar contribution as observed during the previous years. The quantitative contribution of NRW to the soil water balance can become important for crop production, especially under water-limiting conditions, because many ecosystems in Europe were largely affected by
the extreme weather conditions in 2018, which were associated with a decrease in Northern European crop production (i.e. drought, heat wave; Beillouin et al., 2020).

The annual P at the observational site RO was in general lower in the growing seasons (average range: 388 to 569 mm) than during the non-growing seasons (average range: 484 to 581 mm; Table S 3). Similar to GS, the lysimeters from RO in SE
obtained higher P values in growing seasons (average: 402 mm) than in non-growing seasons (average: 213 mm). The transfer of lysimeter brought the soil-ecosystem from a region with high (RO) to a region with low mean P (SE), which consequently led to a reduction of average annual P by 406 mm (i.e., a decline of 40 %) and to a shift in the seasonal P distribution. The latter one is important, as the soil-ecosystems at the two sites are characterized by a different soil water dynamics at the beginning of the growing season. Daily P values in C0CL2 were on average -1.57 and -0.76 mm per day smaller during the
non-growing and growing season, compared to C0CL0 (mean error, Fig. 1 C and D). The annual average NRW was 20 % larger for C0CL0 (55.7 mm) than for C0CL2 (44.4 mm, Table S 4), but the overall contribution of NRW to P was higher for C0CL2 (7.2 %) than for C0CL0 (5.5 %, Table S 5). The seasonal distribution of NRW at RO and SE shows, similar to GS, larger values of NRW during non-growing seasons. The mean daily deviation of NRW between RO and SE was similar during the non-growing and growing season (-0.03 mm). Larger differences were found during winter, when snow covered the
grassland at RO meanwhile in SE, NRW inputs could be determined (Fig. 1 G).
The lowest annual P values of the observation period 2015-2018 were observed in the drought year 2018 with 872 mm at RO and 485 mm at SE (Table S 3). Similarly, also relatively small NRW values could be observed in 2018 at both sites (RO: 45 mm; SE: 38 mm). Despite the lower NRW inputs in 2018, the contribution of NRW to P achieved 7.9 % at SE, which is the highest value during the observation period. Again, this demonstrates the importance of NRW inputs for soil-ecosystem in
dry years when P in form of rainfall P is low.

A direct comparison of P between the manipulative and observational approach was difficult, because only the observational approach includes explicitly a change of P within its design. However, including a change of P, especially during the growing season, is important for climate change studies as the inter-annual variability of the ecosystem productivity (i.e. AGB) are
strongly correlated (Knapp et al., 2018). Hence, climate change experiments that contain information how altered P patterns and droughts affect ecosystem response (i.e. $ET_a$, AGB) would be helpful to further understand how ecosystems will react under conditions with less plant-available soil moisture. However, despite no active control on P within the manipulative approach, treatments with an increased surface temperature affected not only the formation of NRW but also the amount of P.





Here, the effect of local warming on the relative humidity and temperature of the air within and above the canopy could play
an important role, as these influence the vapour pressure gradient between the air inside the leaf and the canopy, which in turn
affects evaporation (soil and surface) and transpiration. At least for NRW, both climate change experiments found similar
patterns and could show that NRW inputs decline when accounting for an increase in temperature (Table 2; Fig.1 E to H), but
their relative contribution to P was only larger under the observational approach (Table S 5). This decline of NRW agrees well
with Tomaszkiewicz et al. (2016), who predicted a decline in dew formation for forecasted trends of increasing temperature
and relative humidity under future climatic scenarios for the Mediterranean region.

### 3.1.2 Evapotranspiration

The average annual $ET_a$ for the manipulative approach at GS ranged between 657 and 831 mm among the different treatments
and reference during the observation period (Table 2, more details in Table S 6). The average difference between the treatments
in the non-growing season were relatively small (max. 24 mm per period) in comparison to the growing season (max. 155 mm
per period; Table 2; Table S 6). During the time with an active control on $CO_2$ and surface temperature, treatments largely
differed from the reference observations. The daily $ET_a$ decreased under the treatment with $CO_2$ enrichment (C2T0) by -
0.40 mm and the treatment with an elevated surface temperature (C0T2) increased the daily $ET_a$ on average by 0.39 mm in
comparison to the reference $ET_a$ (Fig. 1 J). Thus, an elevated $CO_2$ and increased temperature results in contrasting impact on
the $ET_a$ of grassland soil-ecosystems, because of their opposing individual effects on transpiration (Sorokin et al., 2017). Plants
respond to elevated atmospheric $CO_2$ concentrations with a reduced stomatal opening and an increase in photosynthesis (Kruijt
et al., 2008; Ainsworth and Rogers, 2007), which in combination lead to an enhanced plant WUE (Hovenden et al., 2017). In
contrast, elevated temperatures can lead under non-water limiting conditions to an increase in transpiration due to an rise in
vapour pressure deficit and thus enhanced evaporation, because warmer air can retain more moisture (Kirschbaum and
McMillan, 2018); however, it was still unclear which effect has more impact on $ET_a$. Kirschbaum and McMillan (2018)
suggested for a range of locations from tropical to boreal forest that elevated $CO_2$ concentrations had a stronger transpiration-
depressing effect than elevated temperatures. Lenka et al. (2020) showed that treatments with an elevated $CO_2$ concentration
as well as combined increase of $CO_2$ and temperature reduced the stomatal conductance, lowered the $ET_a$, and consequently
improved WUE of soybeans. Our results for the combined treatment C2T2, in general (despite 2018), showed larger daily $ET_a$
values in comparison to the reference, but the average deviation was only 0.12 mm. This result suggests that the effect of
elevated $CO_2$ to reduce $ET_a$ was largely compensated by the effect of elevated temperature in the combined treatment C2T2.
A further comparison of WUE between the treatments could help to clarify which effect was more important for the grassland
ecosystem. Kirschbaum (2004) suggested that the effect of $CO_2$ on $ET_a$ is more pronounced for C3 plants under water limited
conditions and at higher temperatures (Kirschbaum, 2004), because it enhanced the WUE of the plant (Kirschbaum and
McMillan, 2018). However, its effect under non-water limited but temperature limited growth conditions on plants is still
unclear.





As compared to the previous years, the $ET_a$ patterns changed during the growing season of the exceptionally dry and warm year 2018. For C2T0, the growing season $ET_a$ in 2018 showed an increase by 42 mm compared to average values from previous years (2015 to 2017), whereas it declined for C0T2 and C2T2 in comparison to the previous years by 46 and 80 mm,

respectively. This decline suggests that $ET_a$ at the heated plots was temporarily limited by the low availability of water in 2018. In contrast, $ET_a$ of the reference C0T0 did not differ in the growing season 2018 from the average $ET_a$ values of the previous years 2015 to 2017, which demonstrates that the $ET_a$ of soil-ecosystem at the reference plot was not limited by water. Despite the observed increase relative to previous years, C2T0 still showed the lowest $ET_a$ rates of the treatments in 2018. This lower $ET_a$ indicates that an elevated $CO_2$ can mitigate effects of summer droughts on alpine grasslands, which agrees well with

Inauen et al. (2013). They showed that elevated $CO_2$ reduced $ET_a$ by up to 7 % across a range of different grassland types in the central Swiss Alps.

The annual average $ET_a$ of the observational approach was lower at RO (639 mm) than at SE (685 mm; Table 2). The length of the growing season increased on average by 36 days over the observation period due to the translocation of the grassland

soil-ecosystem from RO (C0T0) to SE (C0CL2). Compared with the reference C0CL0, the average daily $ET_a$ of C0CL2 was lower by 0.03 mm in the growing season and larger by 0.33 mm in the non-growing season (Fig. 1 K-L). This reduced $ET_a$ rate is in contrast to findings from the manipulative approach (Fig. 1 J), were treatments with an elevated temperature increased $ET_a$ in comparison to the reference. Main reason for this contrasting response among the different climate change approaches is that the grassland soil-ecosystem from the observational approach was shifted from a rather energy limited to a water limited

regime due to the transfer from RO to SE (Rahmati et al., 2020), whereas the regime at GS is mainly energy limited. Interestingly, during the 2018 European drought-affected growing season, grassland responded with an increase of 4% for C0CL0 in terms of daily $ET_a$, while the daily average $ET_a$ of C0CL2 decreased by 18% in 2018 compared to $ET_a$ of previous years. A similar reduction of $ET_a$ across different ecosystem types could be observed for the sites in Europe affected by this drought in 2018 (i.e. forests, grasslands, croplands and peatlands; Graf et al., 2020). Our results reveal that the limitation in

plant available water at site SE increased during the drought and the heat wave intensified the water stress and thus significantly reduced grassland $ET_a$.

### 3.1.3 Seepage water

The values for the measured water flux across the lysimeter bottom (NetQ) of C2T0 are larger as compared to the reference C0T0 at GS (Figs. 1 M and 1 N). Especially during the growing seasons, the seepage water of C2T0 was during the growing

season 2015 up to 143 % higher than that of C0T0 (Table S 7). Elevated $CO_2$ levels (i.e. C2T0) appear to affect the observed seasonality of seepage through their positive impact on $ET_a$ (reduction), as the differences for NetQ between the growing and non-growing seasons are relatively low compared to all treatments or reference. Vice versa, the treatments C0T2 and C2T2 achieved seasonally lower values of seepage water than C0T0 and thus clearly negative mean errors ($< -0.34$ mm d$^{-1}$; Fig. 1 M and N) during both periods. This demonstrates that water savings due to an elevated $CO_2$ resulted in a significant increase of





seepage water (23 %), whereas treatments with higher temperatures, i.e. C0T2 (-27 %) and C2T2 (-23 %), clearly reduced the
seepage water seasonally up to 69 % in comparison to the reference (Table S 7). Previous investigation for a grassland at the
Swiss Alps showed only a slight increase of seepage water under elevated atmospheric $CO_2$ (Inauen et al., 2013). These
changes in seasonal seepage water could affect the catchment runoff, which is important for hydropower productivity and
profitability in Alpine regions (Anghileri et al., 2018). The European drought in 2018 was clearly visible as the measured NetQ

of all observations during the growing season was reduced to almost zero. Comparing the growing season NetQ in 2018 with
the average NetQ values of the previous years, a severe decline of NetQ, ranging between 95 to 98 % over all observations
was found. This suggests that the groundwater recharge at such alpine grassland sites was considerable affected by the 2018
drought.

The NetQ from the observational approach showed extreme differences between the seasonal values over the observation
period. Largest difference in NetQ between C0CL0 and C0CL2 were observed especially during the non-growing season
(mean error of -2.48 mm; Fig, 1 O). The NetQ of C0CL2 changed in comparison to the reference C0CL0 on average by -
106 %. This demonstrates for the water limited site at SE that the rewetting in the non-growing season is not sufficient to
contribute to NetQ and hence water from P and NRW are mainly used to refill the depleted soil-water storage. Values of NetQ

during the growing season were always negative for C0CL2, which emphasizes an upward directed water flow from deeper
soil layer or groundwater (Table S 7). The severe decline in groundwater recharge during the drought 2018, especially the
NetQ decline during the growing season by 159 % at RO, will eventually also affect stream waters. Although stream flows are
partially buffered by groundwater, the reduced recharge will deplete regional water storage reserves as shown by Fennell et al.
(2020) for a catchment in 2018 in Scotland.

During the drought in 2018, RO showed, similar to GS, a significant decrease in NetQ. In contrast, the soil-ecosystem in SE
showed only low impact of the dry conditions in 2018 on NetQ (-20 mm) compared to observations from the previous years.
This was mainly related to large amounts of P during the autumn and winter months of 2017, which caused a much faster refill
of the soil-water storage in comparison to other years and consequently a larger NetQ that compensated the larger upward
directed water flux at the site during the drought in 2018. Long-lasting droughts can affect the soil-water storage even in the

following year, if winter P is not able to fully replenish the depleted soil water storage (Riedel and Weber, 2020). Thus, we
expect the largest effect on NetQ of the 2018 drought at those sites in the following year.

The response of NetQ to changing climatic conditions was similar for all approaches (excluding C2T0; Fig. 1 M to P). The
decrease of NetQ underlines that the physical and biological response of the ecosystem to changing climatic conditions will

control water fluxes over the land surface, as warming of land surface, reduced precipitation, increased $CO_2$ levels and higher
atmospheric demand for ET, will be altered and thus reduce groundwater replenishment in the future.

**Figure 1:** Comparison of all soil water balance components for all different treatments, or sites of the two different approaches (Lysi-T-FACE: C2T0, C0T2, C2T2; TERENO-SOILCan: C0CL2) to ambient reference of the two different approaches (Lysi-T-FACE: C0T0; TERENO-SOILCan: C0CL0). Subplot A, B, C and D show the daily precipitation (P) of the treatments against the reference observation for the non-growing and growing season (2015 to 2018) for the manipulative (A and B) and the observational experiment (C and D). The same is shown for the non-rainfall water (NRW) in subplot E, F, G, and H, for the actual evapotranspiration ($ET_a$) in I, J, K, and L, as well as for





net water flux across the lysimeter bottom (NetQ) in M, N, O and P. Average daily values were obtained from replicate lysimeter of the same treatment or site. The mean error (mm d$^{-1}$) was calculated to express the average deviation between the daily values obtained under
changed conditions and those of the ambient reference.

## 3.2 Hydroclimatological and ecohydrological indicators

### 3.2.1 Aboveground biomass and water use efficiency

At GS annual AGB were largest during the year 2016 with an average value of 845.2 g m$^{-2}$. The lowest AGB production was quantified in 2015 with an average amount of 709.7 g m$^{-2}$. For the years 2017 and 2018 average AGB value of 821.1 g m$^{-2}$ and
795.2 g m$^{-2}$ were obtained (more details of single years in Table S 8). The year-by-year fluctuations of AGB are mainly induced by weather conditions at GS, as the management (cutting, fertilizer) was identical during the different growing seasons (Fig. 2 A). The reference C0T0 showed on average the largest AGB (1092.5 g m$^{-2}$). Regarding the effect of the different treatments on AGB, the treatment C0T2 achieved on average the largest AGB (743.3 g m$^{-2}$) and the treatment C2T2 with 635.0 g m$^{-2}$ on average the lowest annual AGB production. This reveals clearly that a combined treatment with an elevated temperature and
elevated $CO_2$ concentration (C2T2) reduced the grassland productivity on average by 42 % in comparison to the reference C0T0. The AGB of C2T0 (700.5 g m$^{-2}$) was in general also smaller than the AGB of the reference C0T0. The treatment with only an elevated atmospheric $CO_2$ (C2T0) slightly increased the annual AGB in comparison to C0T2 in the dry 2018, whereas for the other years the AGB of C2T0 was lower than that of C0T2 (Fig. 2 A). This reveals that $CO_2$ fertilization effects were only visible at GS under environmental conditions with less P and higher temperature, which agrees well with previous studies
(see e.g. Morgan et al., 2004; Ainsworth and Rogers, 2007). The overall reduction of AGB of all treatments in comparison to the ambient AGB might be related to an increase of heat stress due to the treatments, which is either directly induced by an elevated canopy temperature or in case of elevated $CO_2$ indirectly, because higher $CO_2$ probably reduce evaporative cooling of plants (Obermeier et al., 2018). The results are in contrast to the widely expected positive effects of increasing $CO_2$ on productivity of agricultural land (Amthor, 2001; Degener, 2015; Zheng et al., 2018).
The negative effect of elevated temperature and elevated $CO_2$ concentration on the AGB could also be seen for the WUE (Fig. 2 C). The WUE of C2T2 with 1.0 g m$^{-3}$ was on average smaller than WUE of C0T2, C2T0 and C0T0 (1.1, 1.3 and 1.8 g m$^{-3}$). For all years, the ambient WUE was much larger than the respective WUE of the treatments, which reveals a negative effect of elevated $CO_2$ concentration and elevated temperature on WUE for this grassland ecosystem. However, comparing WUE among the treatment reveals a higher WUE for C2T0, which was shown previously for a range of different agricultural
ecosystems (Nendel et al., 2009; Roy et al., 2016), because the plants could increase their assimilation rates and simultaneously reduce their water loss by decreasing stomatal conductance under an elevated $CO_2$ (Lammertsma et al., 2011). However, higher temperature seems to increase the "non-productive" water losses, as evident from treatment C0T2 or C2T2, which led to less efficient crop water use as compared to the unheated plots (Fig. 2 C). The small difference between WUE of C0T2 and C2T2 suggests that higher temperatures dominated the response of the grassland ecosystem under conditions with elevated





temperature and atmospheric $CO_2$ concentration (no compensation).The largest difference in WUE between C2T2 and C0T0 could be seen in 2016 and 2018, which might be related to increasing heat stress under all treatments (Obermeier et al., 2018).

The observed AGB for the observational approach achieved on average a higher grassland productivity under wetter and colder (C0CL0: 794 g m$^{-2}$) than under a drier and warmer climate (C0CL2: 721 g m$^{-2}$), with exception of 2015 (Fig. 2 B). During this

year, the larger AGB was mainly related to the AGB from the first cut in the season, where the grassland ecosystems of C0CL2 and C0CL0 achieved on average values of 556 and 316 g m$^{-2}$ (Table S 8). The large differences might be related to the combined effect of moderate drought in 2015 (Ionita et al., 2016) with higher temperatures and solar radiation that led to an earlier start of the growing season for C0CL2 at SE, by meanwhile optimal water availability in spring. Values for AGB from the other cuts were on average much larger under colder and wetter than under warmer and drier conditions, despite the in

average 36 days longer growing season at SE for C0CL2. This was especially visible for the last cuts of the season in 2016 and 2018, when AGB was lower for C0CL2 than C0CL0. Heatwaves in 2016 and 2018 and the decrease of growing season P by 36 % in 2018 in comparison to the average values of the previous years reduced drastically the AGB of C0CL2. The grassland of C0CL2 was exposed to an increasing drought intensity during this period and the plants turned slightly dry and brown and were visibly affected by drought stress in summer 2018 (Rahmati et al., 2020).

The WUE at the colder and wetter climate was on average 1.53 g m$^{-3}$ and thus clearly higher than under warm and dry conditions (1.29 g m$^{-3}$, Fig. 2 D). The results showed that decreasing P and increasing demand of the atmosphere for ET$_0$ led to a decrease of the WUE by 22 %. Average WUE of the 2$^{nd}$ and 3$^{rd}$ cut showed values up to 0.63 g m$^{-3}$, which were the largest differences between C0CL0 and C0CL2, meanwhile WUE of the 1$^{st}$ cut was on average rather similar (data not shown). This suggests that the difference of water availability between C0CL0 and C0CL2 increases within the growing season, which

affects plant growth. The exceptional drought in 2018 affected largely the WUE, as the last cut in 2018 was vanishingly small (0.16 g m$^{-3}$) and decreased by 65 % in comparison to results from C0CL0. The response to drought can be explained for semi-arid to humid, as opposed to arid ecosystems, with a higher sensitivity of ecosystem processes to changes in hydroclimatic conditions (Yang et al., 2016). Niu et al. (2011) showed for a semi-arid grassland from the temperate steppe in Northern China that an increase in P stimulated the WUE of the ecosystem. Similar to our finding, De Boeck et al. (2006) showed that warming

of several grasslands in Belgium led to a decrease of biomass production and WUE. But the same study also implies that WUE of individual species was affected differently by warming experiment, which might led to compositional changes in the ecosystem due to shifts in the competitiveness under altered climate conditions.

Comparing the results of WUE and AGB between both approaches (Fig. 2) suggests that the impact of drought on the WUE

and AGB of grasslands under ambient conditions was much larger in the lowlands than at higher altitudes. However, both approaches responded to dry conditions with a decline in AGB and WUE, except C2T0 treatments, as here $CO_2$ fertilization effects are critical under such environmental conditions. An elevated heat stress might explain the different response to AGB and WUE within the manipulative experiment, whereby for the observational approach additional water stress due low GSP



was the main driver for changes in AGB and WUE. The response of AGB and WUE on altered climatic conditions from the

manipulative and the observational approach suggests that for temperate humid grassland ecosystems, changes in P affected

AGB and WUE in a similar way as elevated atmospheric $CO_2$ and higher temperatures. Further explorations of the distinct

impact of climate change by manipulative and observational approaches should thus include changes in the P regime in addition

to altered $CO_2$ and temperatures.

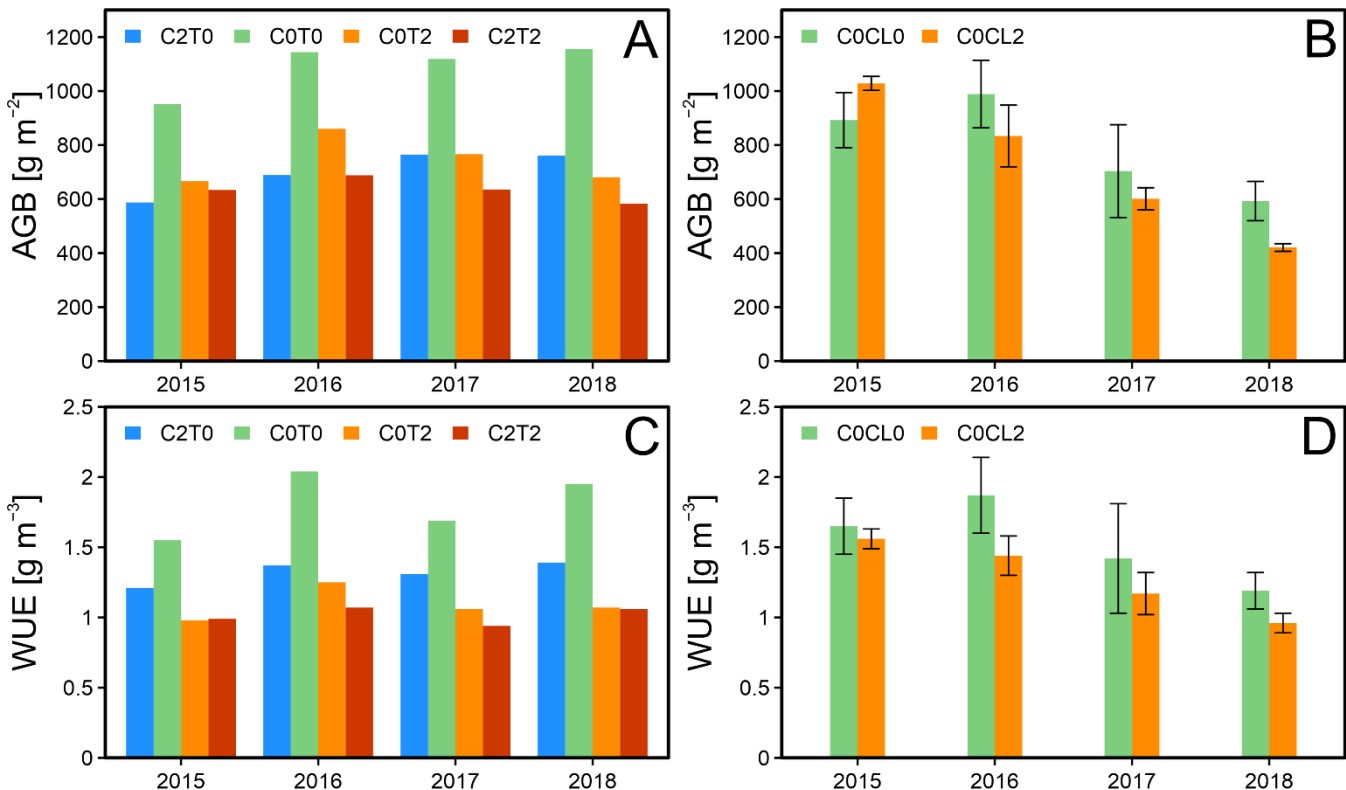

**Figure 2:** Dry aboveground biomass (AGB) during the observational periods 2015-2018 for the manipulative approach Lysi-T-FACE

(C2T0; C0T0; C0T2; C2T2) (A) and observational approach TERENO-SOILCan (Rollesbroich C0CL0; Selhausen C0CL2) (B) as well

crop water use efficiency (WUE) for the manipulative (C) and observational approach (D).

### 3.2.2 Precipitation use efficiency and seepage water to precipitation ratio

The PUE, which includes P as well as NRW, ranged for the manipulative approach Lysi-T-FACE from 0.5 to 1.1 g m$^{-2}$ mm$^{-1}$

across the reference and the different treatments (Fig. 3 A). The highest PUE value was achieved under ambient conditions

(C0T0) in the dry year 2018. All treatments led to a decline of PUE in comparison to C0T0, and showed on average no

differences between each other (0.61 g m$^{-2}$ mm$^{-1}$). The PUE observations suggests that both elevated $CO_2$ and elevated

temperature had a strong effect on the PUE of the grassland ecosystems. However, PUE in dry year 2018 was for C2T0





treatment much larger than C0T2 or C2T2, which might be in addition to the $CO_2$ fertilization effect for AGB, benefit from the higher NRW formation. The formation of NRW is beneficial for plants and their biomass production, because NRW can be either directly taken up by plants (Berry et al., 2019) or indirectly improve their growth conditions (Dawson and Goldsmith, 2018) by e.g. reducing leaf temperature, increasing the albedo, and decrease the vapour pressure deficit (Gerlein-Safdi et al., 2018).

Results from the observational approach showed different patterns under warmer and drier climate conditions. The PUE at C0CL2 was on average 45 % higher than under C0CL0. Nevertheless, Figure 3 B showed also a clear decline of PUE from 2015 (1.37) to 2018 (0.8), which is accompanied by a decrease of P and NRW by 31 and 22 %, respectively, at the site. A similar relationship between the decrease in PUE and the decrease in annual P was found by Jia et al. (2015) for temperate grassland in the loess plateau of China. The lower response of PUE to a decline in the annual P and NRW (Fig. 3; Table S 3 
and S 4) of C0CL0 might be related to the wetter conditions at RO than at SE. The response of the grassland ecosystem from C0CL0 to changing P and NRW is larger in comparison to responses of C0T0 to atmospheric $CO_2$ or surface temperature from the manipulative approach. The elevated warming affects the leaf to air vapour pressure deficit at SE and at the treatments with an elevated temperature at GS. Thus, when the temperature of the leaf surface is above that of the air, the gradient of the vapour pressure will also increase and enhance foliar uptake of water in case of wet leaves (Berry et al., 2019).


The QP ratio obtained for both approaches (Fig. 3 C and D) showed a clear decline if the grassland ecosystem was exposed to warming (i.e. C0T2 and C0CL2). An elevated $CO_2$ increased the QP ratio, as the values were on average 21 % higher than for the reference C0T0. However, the most drastic changes in the QP ratio were visible from the observational approach where QP changed on average by 81 % due to the transfer to a warmer and drier climate (Fig. 3 D). The increase of the QP ratio for 
C0CL2 in 2018 in comparison to the other years is related to the large amounts of P during the autumn and winter months of 2017.



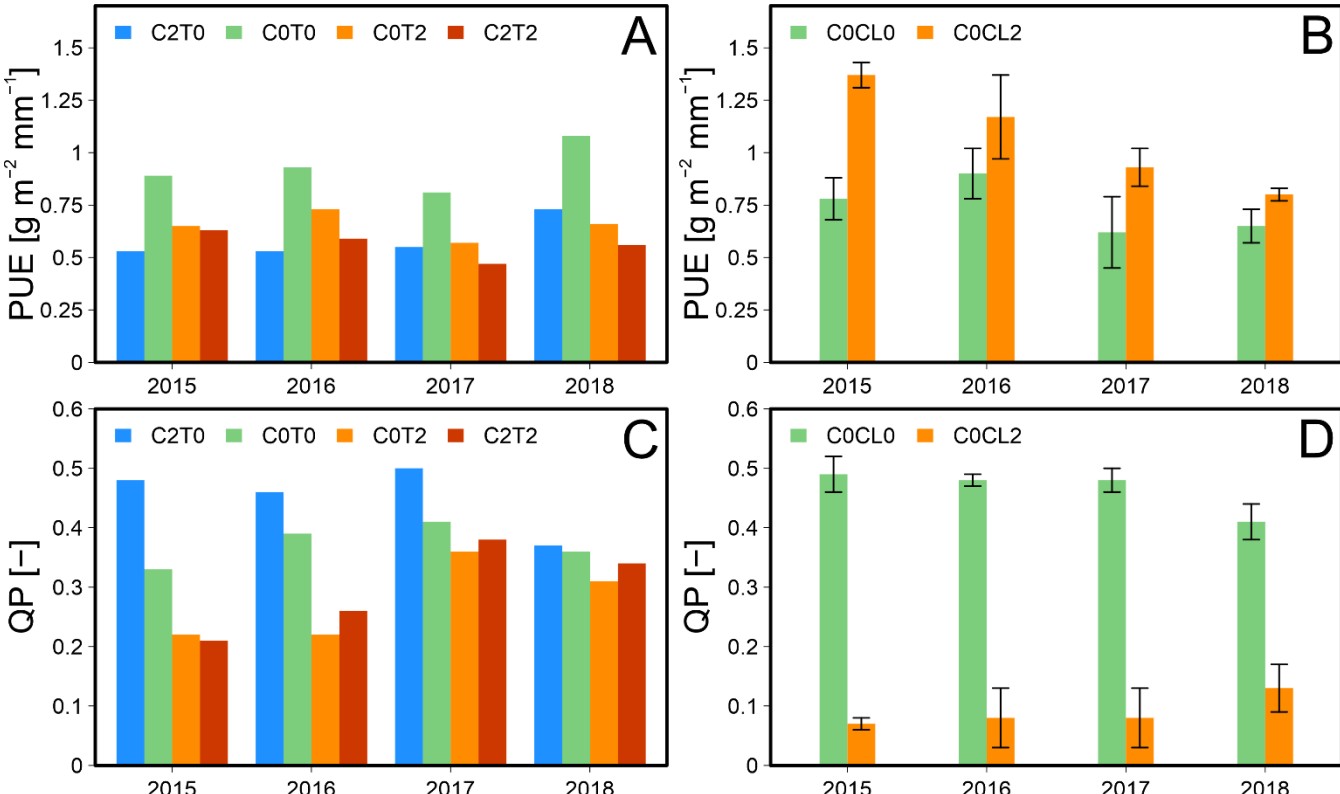

**Figure 3:** Precipitation use efficiency (PUE) during the observational periods 2015-2018 for the manipulative approach Lysi-T-FACE
(C2T0; C0T0; C0T2; C2T2) (A) and observational approach TERENO-SOILCan (Rollesbroich C0CL0; Selhausen C0CL2) (B) as well as
seepage water to precipitation and NRW ratio QP (-) for the manipulative (C) and observational approach (D)

### 3.2.3 Relationship between hydroclimatological and ecohydrological indicators

The ecosystem productivity (i.e. AGB) increased in general with the GSP (Fig. 4 A), and thus the response of the ecosystem
in terms of AGB on GSP has a similar pattern in the distribution between the manipulative and the observational approach.
However, only the relationships from the observational approach were significant (p < 0.01), which suggests the existent of
trade-offs under different P (Wang et al., 2019) and NRW conditions. $ET_a$ in the growing season also increased with increasing
GSP for both approaches, but this relationship was only statistically significant for the grassland of the observational approach
(Fig. 4 B). The low inter-annual variability in growing season $ET_a$ between years with contrasting GSP, e.g. average reduction
of the GSP by 32 % from 2017 to 2018, indicate that water availability is not the limiting factor for $ET_a$ at the alpine site. Our
results agree well with a previous study by Wieser et al. (2008), which showed for 16 sites across different altitudes (580 and
2550 m a.s.l.) that grassland ecosystems in the Austrian Alps seem not to suffer from water stress even in drier years. In
contrast, $ET_a$ from the observational approach showed, especially for the drier and warmer site (C0CL2) with an adjusted $R^2$
of 0.9, a strong correlation link between $ET_a$ and P and NRW during the growing season. The reason for the different response





of the grassland can be explained by the site conditions, as $ET_a$ of the alpine grasslands increased and the $ET_a$ of the low
mountain range grasslands decreased with larger $ET_0$ during the growing season (Figure 4 C). The relationship between AGB
and AI were similar for the observational approach and AGB increased significantly with smaller AI values, when water
limiting conditions diminish according to the Budyko framework (AI<1) (Fig. 4 D); but for the manipulative approach, no
clear changes of AGB with changing AI were visible because of its relatively wet conditions (low AI). The relationship
between WUE and AI achieved similar patterns as shown before for AGB and AI (Fig. 4 E). No relationships from the
manipulative climate change experiment were significant, as inter-annual variability of climate conditions (wet and dry) were
relatively well buffered due to the large availability of water. The AGB and WUE of the grassland-ecosystem from the
observational approach declines significantly with an increasing AI, which suggests that AI can be used as proxy for the site-
specific water availability conditions. In contrast, the alpine grassland showed hardly any change in the above-mentioned
variables on AI. Our results suggest that ecosystem services (e.g. water, yield) of grasslands could differently benefit or suffer
from changing climate conditions. The relationships between the PUE and AI (Fig. 4 F) show that at least for the observational
approach, the PUE of grassland ecosystems adapts to conditions with an increasingly limited water availability.

Altogether, relationships between the AI and AGB and other indicators (WUE and PUE) were different when the grassland
ecosystem was energy-limited (GS and RO) or water-limited (SE). The different treatments of the manipulative approach at
GS did not lead to large shifts in these relationships. More intense changes in the climatic conditions e.g. using rain shelter
might be useful for the manipulative approach to further increase the difference in boundary conditions between treatments
and thus better capture the response of the ecosystem on future climate change. However, results for the observational approach
demonstrate a significant shift (C0CL0 $R^2$ < C0CL2 $R^2$) in the grassland-ecosystem response under climate conditions with a
more pronounced water limitation. However, changes in the soil properties after abrupt changes due to organic matter
decomposition and soil structural changes may have additional effects (Robinson et al., 2016). This highlights that climate
change experiments should include sufficiently extreme conditions to better resolve the key question: "How changes in the
climate regime will affect and alter the ecosystem function in the future?" (Knapp et al., 2018), because modelling the
ecosystem response to changes e.g. like P is of crucial importance to predict future carbon and water cycle (Paschalis et al.,
2020).



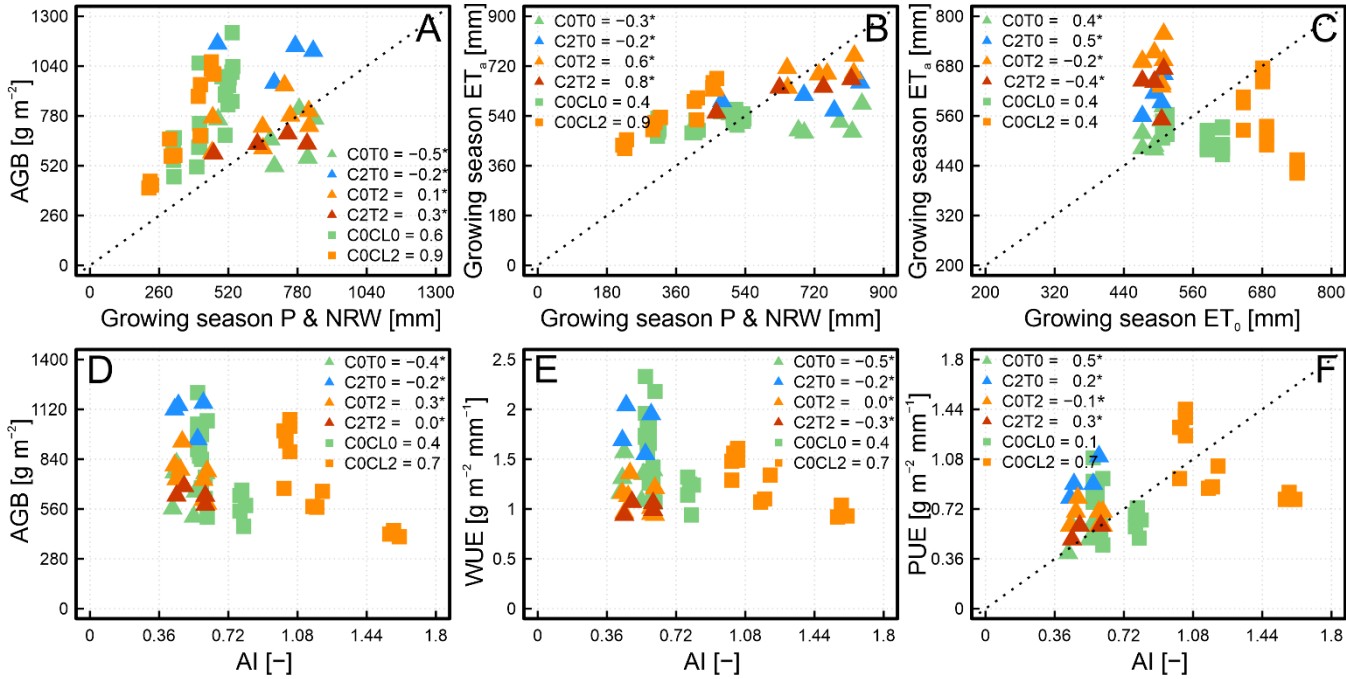


**Figure 4:** Scatterplots of above ground biomass (AGB) to growing season precipitation (P) and non-rainfall water (NRW; A); growing season evapotranspiration (ET$_a$) to growing season P and NRW (B); growing season ET$_a$ to growing season grass-reference evapotranspiration (ET$_0$; C); AGB to aridity index (AI; D); water use efficiency (WUE) to AI (E); and precipitation use efficiency (PUE) to AI (F). In each subplot variables are shown for all treatments from the manipulative (Lysi-T-FACE C2T0; C0T0; C0T2; and C2T2) and observational climate change approaches (TERENO-SOILCan Rollesbroich C0CL0; Selhausen C0CL2) for the period 2015 to 2018.

## 4. Conclusions

A manipulative and an observational approach to assessing climate change impacts on grassland soil ecosystem were analysed and compared over a period of four years, including the exceptionally dry year 2018. The aim was to quantify the effect of changing climatic conditions on the water cycle and productivity of two grassland ecosystems by the use of high precision lysimeter data. The main hypothesis was that the response of the grassland ecosystems on changing climatic conditions will differ among the two distinct experimental approaches.

Results of the comparisons between grassland ecosystems from two different climate change approaches suggest that the most critical factor was the plant water availability. Precipitation (P) and non-rainfall water (NRW) amount and temporal variability of the events played a major role on the response of the ecosystem to future climate, but response in terms of the water balance and biomass production are partially buffered by the soil i.e. that water or heat stress can be mitigated by the status of ecosystem conditions. Thus, climate change experiments that contain information how altered P and NRW patterns and droughts affect





ecosystem response (i.e. actual evapotranspiration-$ET_a$, above ground biomass-AGB) would be helpful to further understand how ecosystems will react under soil conditions with less plant-available water.

Results from the manipulative climate change approach indicating that both elevated temperature and elevated $CO_2$
concentration altered P and NRW suggest that effects on NRW and P need to be considered in global change experiments to avoid an over- (elevated temperature) or underestimation (elevated $CO_2$) of the effects of climate change on ecosystems response, especially for sites where water limitation plays a role. The elevated temperature was the dominant factor for changing $ET_a$ of the grassland ecosystem under non-water limited but temperature limited growth conditions, as effect of elevated $CO_2$ concentration on $ET_a$ was largely compensated by higher temperatures (increasing $ET_a$) within the combined
treatment. Future climate change experiments should account for these effects and thus should include elevated $CO_2$ concentrations, especially under drier environmental conditions or droughts where such a compensation is most decisive, because water and heat stress for plants significantly affects productivity (e.g. AGB) and the water use efficiency (WUE) of grassland ecosystem. The modification of the seasonal patterns of groundwater recharge from both approaches depends on the response of plant and soil ($ET_a$) to changing climatic conditions, suggesting that changes affecting water fluxes from the land
surface will play a crucial role to predict future ecosystem water balances. The effects of drought on seepage water that potentially leading to groundwater recharge were more pronounced at sites under wetter soil conditions, because here the higher demand for evapotranspiration can be satisfied on the expense of a decreasing soil water storage. The response of AGB and WUE on altered climatic conditions from the manipulative and the observational approach suggests that changes in boundary conditions (i.e. P, elevated $CO_2$ or temperature) affects both variables in temperate humid grassland ecosystem in a
similar way. Indicators such as aridity index and its relationship to biomass production and plant water use revealed that the status of the grassland ecosystem is important (i.e., temperature, energy, or water limited) to understand response of ecosystem to changing climatic conditions. The inter-annual variability of climatic conditions in the context of climate change experiments was found crucially important for the water budget and the connected aboveground biomass production, since the more different the conditions, the clearer the effects, including the occurrence of temperature or water stress, the grassland
ecosystem seems to show a clear response.

The results of this study confirmed that both climate change approaches, depend mainly on the ecosystems' status with respect to temperature and energy or water limitation. The larger impact of drought on biomass production in wetter climates suggests that ecosystems may adapt upon conditions during drought period.

Thus, climate change experiments should consider both the status of the ecosystem as well as the occurrence of extreme
conditions. Future studies should integrate all factors (precipitation, temperature and $CO_2$) in order to obtain a complete ecosystem response, which may help improving model predictions of how changes in climate regimes will affect the function of ecosystems.

**Data availability**



All data for the specific lysimeter and weather station (raw data) can be freely obtained from the TERENO data portal (https://teodoor.icg.kfa-juelich.de/ddp/index.jsp (last access: 9 February 2021; Kunkel et al., 2013), lysimeter station Rollesbroich and Selhausen: RO_Y_01 and SE_Y_02). The processed data to support the findings of this study can be acquired upon request from Jannis Groh. The raw data for the lysimeter and weather station can be obtained upon request from Markus Herndl.

**Author contribution**

MH and TP conceived the experiments at the corresponding site in Austria and Germany. JG and VF had the idea and designed the study. MH and JG provided the data for the corresponding lysimeter stations. VF and JG performed the data analysis, and reviewed and wrote the manuscript with equal contributions from all co-authors.

**Acknowledgements**

We acknowledge the support of TERENO and SOILCan, which were funded by the Helmholtz Association (HGF) and the Federal Ministry of Education and Research (BMBF). The authors also acknowledge the financial support by the University of Graz. The lysimeter facility and the project 'Lysi-T-FACE' (DaFNE, 100719) at Gumpenstein was funded by the Austrian Federal Ministry of Agriculture, Forestry, Environment and Water Management (BMLFUW). Veronika Forstner is funded by a DOC Fellowship of the Austrian Academy of Sciences (ÖAW). Matevz Vremec is funded by the Earth System Sciences
research program of the ÖAW (project ClimGrassHydro). We thank the colleagues at the corresponding lysimeter station for their kind support: Martina Schink, Matthias Kandolf, Irene Sölkner (Gumpenstein), Werner Küpper, Ferdinand Engels, Philipp Meulendick, Rainer Harms, and Leander Fürst (Selhausen, Rollesbroich).

**Competing interests**

The authors declare that they have no conflict of interests.






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
