# Peer review of "Response of water balance components to climate change in permanent grassland soil ecosystems"

_Hydrology and Earth System Sciences, 2021_

## Author Response (AR1)

**Response to comments by Referee #1**

In this manuscript, Forstner et al examined the response of water balance components to climate change in the grassland using manipulative or observational approaches. The authors did a solid job on data processing and the topic itself is also important. However, base on the current version, I feel this manuscript still needs a better polish on the structure and writing.

Response: We are glad about the positive feedback and thank Referee #1 for the constructive and helpful comments. We carefully revised the structure (i.e. splitting Result and Discussion), improved the writing and readability of the manuscript as well as the general presentation of our findings.

1. My major concern is manuscript writing. Overall, it hard to get the key message of this manuscript. For example, the abstract and conclusion sections are too long and almost the same length as the introduction section. I did not see the difference between the abstract, introduction, and conclusion sections. Too many details in the abstract and conclusion make me feel really difficult to get the key findings and conclusions of this paper.

Response: We revised the abstract and conclusion carefully to improve the readability and hope that key message from our investigation are now clearer. Our key message is:
The response of soil water fluxes and biomass production of both climate change approaches, depend mainly on the ecosystems' status with respect to temperature and energy or water limitation. To thoroughly understand the ecosystem response based on observation from climate change experiments sufficiently extreme boundary conditions as well as important factors such temperature, $CO_2$ and precipitation and NRW should be integrated in the experimental set-up to obtain a complete ecosystem response and better understanding of tipping points in ecosystems

2. In the title "Response of water balance components to climate change in permanent grassland soil ecosystems". My first impression when I saw water balance components, I think AGB is not like ET and WUE which are directly related to the water balance component.

Response: We agree and changed the title to: "Response of water fluxes and biomass production to climate change in permanent grassland soil ecosystems".

3 The results and discussion part is overall too descriptive and lacks in-depth discussion. I expected to read more about the strengths and limitations of the manipulative or observational approaches for estimating the impact of climate change on the ecosystem, and some key guidance on how we could better use those methods in the future. However, it seems this part is still lack in the discussion. I would suggest the authors separate the results and discussion.

Response: We followed the suggestion of the Referee #1 and separated the result and discussion part. In addition, we structured the discussion to improve a better readability of the manuscript and add a more in depth discussion of our findings (see L704 – 893 in the track-changes version). In the discussion, we have further elaborated the strengths and weaknesses of each approach and made recommendations for future experiments based on our findings.

**Response to comments by anonymous referee #2**

Thanks to the authors (Forstner et.al) for their nice work. I hope the provided review is a helpful contribution to the submitted article.
The authors addressed the effects of climate change on water balance and the productivity of the grassland ecosystem by using two different climate change experiment approaches, namely manipulative and observational. Moreover, the authors pointed the effects of the modified climatologic conditions on the relationship between water balance and hydroclimatic and ecohydrological indicators. The authors also included non-rainfall water input into the water balance calculation. Nevertheless, I think some points can be improved.

Response: We are glad about the positive evaluation of our work and thank Referee #2 for the constructive and helpful comments.

**General comments:**

The paper is well structured, yet it is not easy to follow the authors' thoughts. Thus, the article would profit from general prove reading to shorten some sentences. The field setups etc. are already difficult to follow, therefore it might be better to add key findings of the study into short and clear sentences, especially in the conclusion.

Response: We thoroughly revised the manuscript, shortened abstract and conclusion, separated result and discussion section. We hope this helps to improve the readability and clarifies the view on our key findings.

Secondly, please consider deepening the discussion. For example, describing the effect of climate change on soil water availability, and/or soil moisture change. Moreover, a detailed comparison of the approaches about their effects on the water balance components. For example, by addressing how the weaknesses/strengths of the two experimental approaches influence understanding the relationship between climate change and the water balance & ecosystem productivity.

Response: We improved and structured the discussion of the manuscript (see L704 – 893; track-changes version), in particular, with respect to the comparison of the two approaches. We also clarified by an additional figure 2 (see L395; track-changes version) the hydrological status of the ecosystem by the use of the Budyko -framework. We added the water balance component "change in soil water storage" to the investigation (see L469; track-changes version) to be better describe effect of climate change on soil water availability of the entire soil-ecosystem. Within the new discussion sections, we also discuss the weakness and strength of the two experimental approaches (e.g. see L741-743, L777-782, L831-833, L888-893; track-changes version).

**Specific comments:**
**Introduction**

- Line 56: there is a typo error, please change 'biomes' to 'biomass'

Response: We refer here to biome and not to biomass: We changed it to "Grasslands represent one of the main biomes of the earth"

- Line 90, 'these research questions..' is a bit ambiguous description since in the former parts of the introduction the research questions were not clearly stated. The general need for the research and usability of the hydroclimatic and ecohydrological indicators were mentioned. Please consider re-structuring the paragraph which starts with line 90.

Response: We clarified our research question in L126 in the track changes version to: "To identify the response of water fluxes and AGB production to climate change in permanent grassland soil ecosystems two major approaches have been propose. Either manipulative experiments (using multifactorial drivers) or observational experiments on environmental gradients can serve to explore the relationship between changing climate factors and ecosystem responses (Hanson and Walker, 2020; Song et al., 2019; Kreyling and Beier, 2013; Knapp et al., 2018).

- Line 91: '… are one major tool…' please consider to re-write as ' …are major tools' or ' is one of the major tool'

Response: We changed the sentence to: Either manipulative experiments (using multifactorial drivers) or observational experiments on environmental gradients can serve to explore the relationship between changing climate factors and ecosystem responses (Hanson and Walker, 2020; Song et al., 2019; Kreyling and Beier, 2013; Knapp et al., 2018).

- Line 101: 'GSP' abbreviation is not clear, please consider the write the abbreviation just after '…between growing season P' or in the line of 103 after '..growing season precipitation. Since many abbreviations were repeatedly used, it's helpful for the reader to have a clear explanation of each abbreviation.

Response: We changed it in the track changes version in L139 to: …..between growing season precipitation (GSP) and AGB, when…

**Methods:**
It can be nicely followed, and it is a well-written section. But, to clarify the reader, you might consider describing soil water availability. To be sure what the authors mean by 'soil water availability is conveyed to the reader clearly. It might be helpful for the reader to evaluate the water balance components of the sites if you give an average duration of growing period/non-growing period of the plots (GS,RO,SE).

Response: We clarified within the revised manuscript the meaning of soil water availability by adding the water balance component $\Delta S$ (i.e. change in soil water storage) to the analysis. The water availability in an ecosystem can be in general determined by it hydrological status, thus we added a Budyko-framework to the M&M section

(L279-L283) and the Budyko plot (see Fig.2 in L413 of the track changes version), which nicely characterizes between the demand (energy) and the supply limitations of an ecosystem (i.e. water). Thank you for this suggestion, we now also added the following sentences to the Material and Method section to provide the reader the information about the average duration of the growing period:

L181-182: "The grassland was mowed three times per year (see supplement Table S 1), each followed by mineral fertilization (Herndl et al., 2010), and the average length of the growing season over the observation period from 2015 to 2018 was 197 days (more details on the method see chapter 2,3)."

L200-201: Mean length of the growing season over the observation period was 208 and 243 days in RO and SE, respectively.

- Line 138: there is a typo error, please write 'the Alps'

Response: We changed it as suggested

- Line 158: 'Data from these stations…', here "these stations" are not a clear words-group. Is it the weather station and the mechanical snow cover separation system or are there multiple weather stations in the one plot, or does it refer to all weather stations that were used in the study? Please think to restructure the sentence.

Response: Only one station was used for the site GS. We clarified this within the revised version (L192; track change version)

Line 158: please consider splitting the paragraph when you mentioned another experiment setup (here is RO)

Response: We changed it as suggested

- Line 159-160: please think to restructure the sentence since it is difficult to understand

Response: We changed this sentence to: At RO (C0CL0, CL stands for climate) six lysimeters were installed to quantify soil water budget and AGB under ambient conditions (RO: C0CL0). At SE (C0CL2) three lysimeter were installed to quantify soil water fluxes and AGB under altered climatic conditions (SE: C0CL2, less precipitation, higher $ET_0$).

- Line 235: the unit of the aridity index, please check

Response: thanks for this advice. The unit of the AI is dimensionless. We have changed it in the revised version

- Line 245: there is a typo error: please change 'statically' to statistically

Response: Done

**Results and discussion:**
The results were nicely stated. NRW was nicely discussed through the different sites & setups. However, it would enrich the paper if the authors add more opinion on the differences of the approaches for estimating the impact of climate change by commenting on the limitations/advantages of the approaches. Detailed opinion on how climate change influences soil water availability by describing the processes and the experimental approaches' challenges help to deepen the discussion.

Response: We compared and thoroughly discussed the differences between the approaches more in deep within the revised version of the manuscript. Apart from separating the results and discussion section, the new discussion sections, also includes discuss on the weakness and strength of the two experimental approaches (e.g. see L741-743, L777-782, L831-833, L888-893; track-changes version).

- Line 256: ' the lysimeter solely enriched with CO2…' is not a clear sentence, since there are 2 lysimeters for C2T0 ( except the year 2017 & 2018). Therefore, here either the authors referred to the years 2017 and 2018 or it is the average of the two lysimeter data. Please consider re-structure the sentence.

Response: We changed it to: "The lysimeters solely enriched with $CO_2$ … "

---

## Referee Report (RR1)

Dear Authors and Dear Editor,

Thank you for considering my comments on your manuscript and addressing all the points mentioned in the previous review.

I have a minor point remain that should be addressed to improve the manuscript:

In method and discussion section, please consider to check tenses of the sentences to have coherence. For example: in L128 'at GS six lysimeters are installed…'  present tense and in L146 'at RO six lysimeters were installed' past tense were used.

Best regards

---

## Author Response (AR2)

**Response to comments by Referee #2**

Thank you for considering my comments on your manuscript and addressing all the points mentioned in the previous review. I have a minor point remain that should be addressed to improve the manuscript: In method and discussion section, please consider to check tenses of the sentences to have coherence. For example: in L128 'at GS six lysimeters are installed…' present tense and in L146 'at RO six lysimeters were installed' past tense were used.

Response: Thank you for your positive feedback on our revision. Thank you very much for the point about the tenses; we carefully looked at the tenses in the whole manuscript.

L128: We have changed accordingly to: "At GS six lysimeters were installed; one lysimeter was operated …" (track changes version)

**Response to comments by Referee #3**

The two main issues raised by the reviewers were a lack of depth in the discussion and of synthetic writing throughout the manuscript. Both have been appropriately addressed, and the manuscript has been substantially rewritten. This makes the article easier, clearer and more exciting to read than the previous version. All minor comments have been addressed as well. I would therefore definitely recommend this paper for publication in HESS, provided that some last small errors and typos are corrected (please note that line numbers correspond to the author's tracked changes "HESS-2021-100-ATC1.pdf" version):

Response: Thank you for the positive evaluation of our revision. We are glad that now the manuscript is easier to read and that the main highlights are clearer. Thank you for observing the errors in some lines; we corrected the sentences accordingly.

84-85: "as well as precipitation" instead of "as well precipitation"
Response: The changes have been made accordingly in L39 of the track changes version.
186-187: a bit naive question about coding of the treatments: why 0 for ambient and 2 for elevated, and not 0 and 1 instead?
Response: Thanks for this comment. The reason why we used 2 instead of 1 is related to the overall design of the experimental site, as here also other plots (without lysimeter observations) with an elevated $CO_2$ of +150 ppm exist (and are indicated by 1).
191: what does GWU stands for? I don't think this acronym has been defined before, but I may have missed it.
Response: We have deleted GWU (L135, track changes version).
230: there is one "the" too much in this sentence
Response: We have changed it to "The change in soil water storage ($\Delta S$), which affects water availability in the soil ecosystem, […]" see L164 and L165 in the track changes version.
275: maybe "and summed up" instead of "and added"?
Response: We have changed it accordingly to "and summed-up", see L205 in the track changes version
310: why using $p < 0.01$ as significance threshold and not $p < 0.05$? Is that a sort of Bonferroni correction because there many linear models are tested?
Response: We changed it to the more conventionally use p-value of $< 0.05$. This did not change the reported results.
873: "steeper" instead of "stepper"
Response: We correct the word into "steeper" in L628 of the track changes version

---

## Author Response (AR3)

Dear Editor,

we have attached the old manuscript version with all figures, but as requested, we have uploaded high-quality vector-based figures (format pdf and EPS) for the manuscript separately in "Figures (zip)".

We hope that you have now received all the necessary files from us. If you have any further questions, please do not hesitate to contact us.

Yours sincerely, Veronika Forstner and Jannis Groh